# Polygenic risk scores in cardiovascular risk prediction: A cohort study and modelling analyses

Luanluan Sun[1☯], Lisa Pennells[1☯], Stephen Kaptoge[1☯], Christopher P. Nelson[2], Scott C. Ritchie[1,3], Gad Abraham[1,3], Matthew Arnold[1], Steven Bell[1], Thomas Bolton[1], Stephen Burgess[1], Frank Dudbridge[2,4], Qi Guo[1], Eleni Sofianopoulou[1], David Stevens[1], John R. Thompson[2], Adam S. Butterworth[1], Angela Wood[1], John Danesh[1,5‡], Nilesh J. Samani[2,4‡], Michael Inouye[1,3,6‡*], Emanuele Di Angelantonio[1‡*]

1 Department of Public Health and Primary Care, University of Cambridge, Cambridge, United Kingdom, 2 Department of Cardiovascular Sciences and NIHR Leicester Biomedical Research Centre, University of Leicester, Leicester, United Kingdom, 3 Cambridge Baker Systems Genomics Initiative, Baker Heart and Diabetes Institute, Melbourne, Victoria, Australia, 4 Department of Health Sciences, University of Leicester, Leicester, United Kingdom, 5 Wellcome Trust Sanger Institute, Wellcome Genome Campus, Hinxton, United Kingdom, 6 The Alan Turing Institute, London, United Kingdom

☯ These authors contributed equally to this work.
‡ These authors are joint senior authors on this work.
* mi336@medschl.cam.ac.uk (MI); ed303@medschl.cam.ac.uk (EDA)

**Data Availability Statement:** All data files are available from the UK Biobank and CPRD databases.

# Abstract

## Background

Polygenic risk scores (PRSs) can stratify populations into cardiovascular disease (CVD) risk groups. We aimed to quantify the potential advantage of adding information on PRSs to conventional risk factors in the primary prevention of CVD.

## Methods and findings

Using data from UK Biobank on 306,654 individuals without a history of CVD and not on lipid-lowering treatments (mean age [SD]: 56.0 [8.0] years; females: 57%; median follow-up: 8.1 years), we calculated measures of risk discrimination and reclassification upon addition of PRSs to risk factors in a conventional risk prediction model (i.e., age, sex, systolic blood pressure, smoking status, history of diabetes, and total and high-density lipoprotein cholesterol). We then modelled the implications of initiating guideline-recommended statin therapy in a primary care setting using incidence rates from 2.1 million individuals from the Clinical Practice Research Datalink. The C-index, a measure of risk discrimination, was 0.710 (95% CI 0.703–0.717) for a CVD prediction model containing conventional risk predictors alone. Addition of information on PRSs increased the C-index by 0.012 (95% CI 0.009–0.015), and resulted in continuous net reclassification improvements of about 10% and 12% in cases and non-cases, respectively. If a PRS were assessed in the entire UK primary care population aged 40–75 years, assuming that statin therapy would be initiated in accordance with the UK National Institute for Health and Care Excellence guidelines (i.e., for persons with a

**Funding:** This work was supported by core funding from the UK Medical Research Council (MR/L003120/1), the British Heart Foundation (RG/13/13/30194; RG/18/13/33946), and the National Institute for Health Research (NIHR) (Cambridge Biomedical Research Centre at the Cambridge University Hospitals NHS Foundation Trust and NIHR Leicester Biomedical Research Centre). This work was supported by Health Data Research UK, which is funded by the the UK Medical Research Council, the Engineering and Physical Sciences Research Council, the Economic and Social Research Council, the Department of Health and Social Care (England), the Chief Scientist Office of the Scottish Government Health and Social Care Directorates, the Health and Social Care Research and Development Division (Welsh Government), the Public Health Agency (Northern Ireland), the British Heart Foundation, and Wellcome. Luanluan Sun, Lisa Pennells, Stephen Kaptoge, and Matthew Arnold are funded by a British Heart Foundation Programme Grant (RG/18/13/33946). Christopher P. Nelson is funded by a British Heart Foundation Grant (SP/16/4/32697). Scott Ritchie, Mike Inouye, and Stephen Burgess are funded by the National Institute for Health Research (Cambridge Biomedical Research Centre at the Cambridge University Hospitals NHS Foundation Trust). David Stevens was funded by the National Institute for Health Research (Cambridge Biomedical Research Centre at the Cambridge University Hospitals NHS Foundation Trust). Thomas Bolton is funded by the NIHR Blood and Transplant Research Unit in Donor Health and Genomics (NIHR BTRU-2014-10024). Steven Bell was funded by the NIHR Blood and Transplant Research Unit in Donor Health and Genomics (NIHR BTRU-2014-10024). Angela Wood is supported by a BHF-Turing Cardiovascular Data Science Award and by the EC-Innovative Medicines Initiative (BigData@Heart). Professor John Danesh holds a British Heart Foundation Professorship and a National Institute for Health Research Senior Investigator Award.

**Competing interests:** I have read the journal's policy and the authors of this manuscript have the following competing interests: SK is funded by grants to institution from: British Heart Foundation, UK Medical Research Council, UK National Institute of Health Research, Cambridge Biomedical Research Centre. SB is a paid statistical reviewer for PLOS Medicine. ASB received grants outside of this work from AstraZeneca, Biogen, Bioverativ, Merck, Novartis and Sanofi, as well as personal fees from Novartis. JD serves on the International Cardiovascular and Metabolic Advisory Board for Novartis (since 2010), the Steering Committee of UK Biobank (since 2011), the MRC International

predicted risk of ≥10% and for those with certain other risk factors, such as diabetes, irrespective of their 10-year predicted risk), then it could help prevent 1 additional CVD event for approximately every 5,750 individuals screened. By contrast, targeted assessment only among people at intermediate (i.e., 5% to <10%) 10-year CVD risk could help prevent 1 additional CVD event for approximately every 340 individuals screened. Such a targeted strategy could help prevent 7% more CVD events than conventional risk prediction alone. Potential gains afforded by assessment of PRSs on top of conventional risk factors would be about 1.5-fold greater than those provided by assessment of C-reactive protein, a plasma biomarker included in some risk prediction guidelines. Potential limitations of this study include its restriction to European ancestry participants and a lack of health economic evaluation.

## Conclusions

Our results suggest that addition of PRSs to conventional risk factors can modestly enhance prediction of first-onset CVD and could translate into population health benefits if used at scale.

## Author summary

### Why was this study done?

- Application of polygenic risk scores (PRSs) has opened opportunities to enhance risk stratification and prevention for common diseases. The clinical utility of PRSs in cardiovascular disease (CVD) risk prediction is, however, uncertain.

- Previous analyses have generally focused only on coronary heart disease (CHD) rather than the composite outcome of CHD and stroke, and have often lacked modelling of clinical implications of initiating guideline-recommended interventions (e.g., statin therapy).

### What did the researchers do and find?

- We quantified the incremental predictive gain with PRSs on top of conventional risk factors using data on 306,654 individuals from UK Biobank.

- We modelled the population health implications of initiating statin therapy as recommended by current guidelines using data from 2.1 million individuals from the Clinical Practice Research Datalink.

- Addition of information on PRSs to a conventional risk prediction model increased the C-index (a measure of risk discrimination) and improved risk classification of cases and non-cases.

- We estimated that targeted assessment of PRSs among people at intermediate (i.e., 5% to <10%) 10-year CVD risk could help prevent 1 additional CVD event for

Advisory Group (ING) member, London (since 2013), the MRC High Throughput Science 'Omics Panel Member, London (since 2013), the Scientific Advisory Committee for Sanofi (since 2013), the International Cardiovascular and Metabolism Research and Development Portfolio Committee for Novartis and the Astra Zeneca Genomics Advisory Board (2018).

**Abbreviations:** CABG, coronary artery bypass grafting; CHD, coronary heart disease; CPRD, Clinical Practice Research Datalink; CRP, C-reactive protein; CVD, cardiovascular disease; GWAS, genome-wide association study; HDL, high-density lipoprotein; HES, Hospital Episode Statistics; LDL, low-density lipoprotein; NRI, net reclassification index; PRS, polygenic risk score; PTCA, percutaneous transluminal coronary angioplasty; UKB, UK Biobank.

approximately every 340 individuals screened, which would be almost 15 times more efficient than blanket assessment of PRS.

## What do these findings mean?

- Addition of PRSs to conventional risk factors provided modest improvement in prediction of first-onset CVD.

- Nevertheless, these moderate improvements could translate into meaningful clinical benefit if applied at scale, and lead to the prevention of 7% more CVD events than conventional risk factors alone.

- Our results also suggest that targeted use of PRSs would be more efficient than blanket population-wide use.

- Future studies should seek to evaluate PRSs in non-European ancestry populations, and perform formal health economic evaluations.

## Introduction

Advances in the application of polygenic risk scores (PRSs) have opened opportunities to enhance disease risk prediction by stratifying populations into risk groups using information on millions of variants across the genome [1–4]. The UK government's Department of Health and Social Care green paper on disease prevention has stated: 'As the evidence develops, complementing existing risk scores. . .with this kind of genetic information [i.e., PRSs] will be a priority for the UK healthcare system' [5]. The US Centers for Disease Control and Prevention and the US National Institutes of Health are also considering the value of integrating PRSs into clinical practice [6].

A key strategy in the primary prevention of cardiovascular disease (CVD) is the use of risk prediction algorithms to target preventive interventions to people who may benefit from them most [7–12]. These algorithms typically include information on conventional risk factors, including age, sex, smoking history, history of diabetes, blood pressure, total cholesterol, and high-density lipoprotein (HDL) cholesterol [8–10]. The population health utility of PRSs in CVD risk prediction is, however, uncertain. Previous analyses have generally focused only on coronary heart disease (CHD) rather than the composite outcome of CHD and stroke, even though the composite outcome is the focus of most primary prevention guidelines. Furthermore, most previous PRS studies have lacked modelling of the clinical implications of initiating guideline-recommended interventions (e.g., statin therapy) [13,14], meaning that it has been difficult to judge the potential clinical gains of assessing PRSs.

Our study, therefore, aimed to address 2 questions. First, what is the improvement in CVD risk prediction when PRSs are added to risk factors used in conventional risk algorithms? We analysed 306,654 participants from UK Biobank (UKB) to assess the value of adding PRSs to several conventional risk factors. Second, what is the estimated population health impact of using information on PRSs for CVD prediction? We modelled data from 2.1 million individuals in the Clinical Practice Research Datalink (CPRD) to estimate the benefit of initiating statin therapy as recommended by guidelines. To contextualise our findings, we compared the

incremental predictive gains afforded by PRSs with that provided by C-reactive protein (CRP), a plasma biomarker recommended for risk prediction in some CVD primary prevention guidelines [12,15].

## Methods

### Study design and overview

Our study involved several interrelated components (Fig 1). First, we constructed separate PRSs for CHD and stroke, using methods previously described [16,17]. Second, we calculated measures of risk discrimination and reclassification to quantify the incremental predictive gain with these PRSs on top of conventional risk factors. Third, to estimate the potential for disease prevention in a general population setting, we adapted (i.e., recalibrated) our findings to the context of a primary prevention population eligible for CVD screening, using incidence rates from contemporary computerised records from general practices in the UK. Fourth, we modelled the clinical implications of initiating statin therapy as recommended by current guidelines, comparing a 'blanket' approach (i.e., assessment of PRSs in all individuals eligible for CVD primary prevention) with a 'targeted' approach (i.e., focusing PRSs assessment only in people judged to be at intermediate 10-year risk of CVD after initial screening with conventional risk predictors alone). Fifth, to help contextualise the potential population health gains

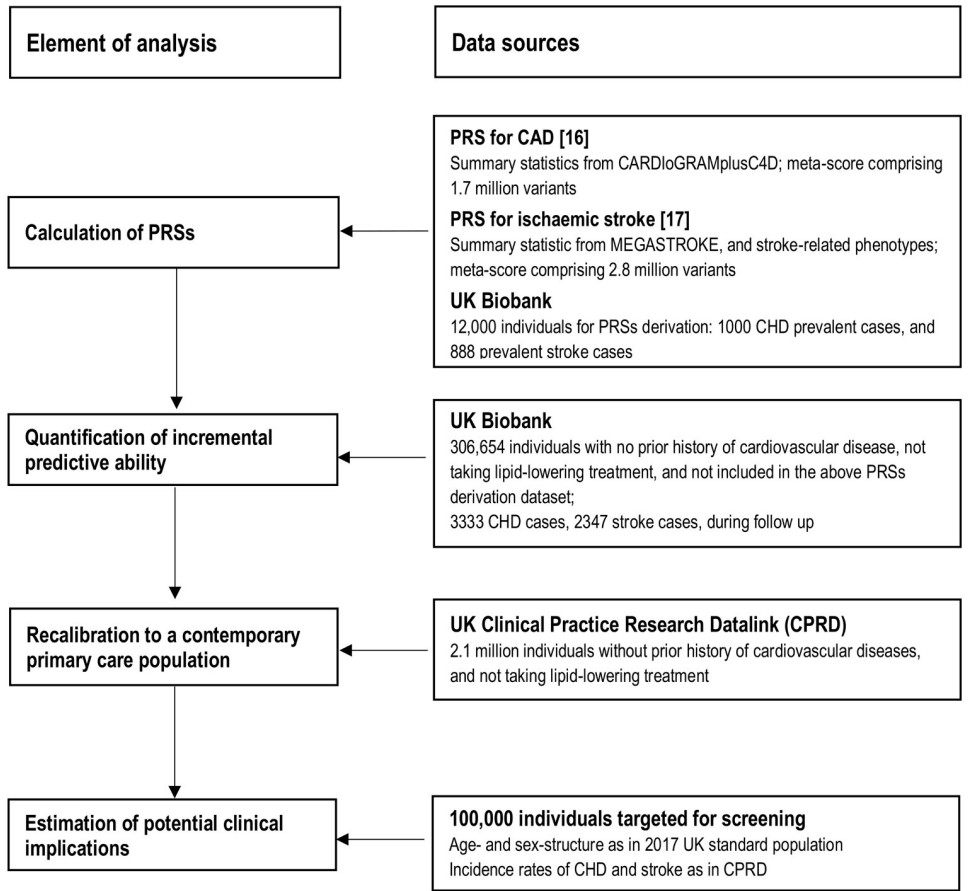

**Fig 1. Study design and overview.** CHD, coronary heart disease; PRS, polygenic risk score.

afforded by assessing PRSs, we compared them in the same dataset with gains afforded by assessment of CRP.

## Ethics statement

This research has been conducted using the UKB resource under application number 26865. The UKB study was approved by the North West Multi-centre Research Ethics Committee, and all participants provided written informed consent to participate in the UKB. This study is based in part on data from the CPRD obtained under licence from the UK Medicines and Healthcare products Regulatory Agency (protocol number 162RMn2). The data are provided by patients and collected by the National Health Service as part of their care and support.

## Data sources

**UK Biobank prospective study.**   Details of the design, methods, and participants of UKB have been described previously [18,19]. Briefly, participants aged 40 to 75 years identified through primary care lists were recruited across 22 assessment centres throughout the UK between 2006 and 2010. At recruitment, information was collected via a standardised questionnaire and selected physical measurements. Details of the data used from UKB are provided in S1 Text. Data were subsequently linked to Hospital Episode Statistics (HES), as well as national death and cancer registries. HES uses the International Classification of Diseases (ICD) 9th and 10th revisions to record diagnosis information, and the Office of Population, Censuses and Surveys: Classification of Interventions and Procedures, version 4 (OPCS-4), to code operative procedures. Death registries include deaths in the UK, with both primary and contributory causes of death coded according to ICD-10.

Genotyping was undertaken using a custom-built genome-wide array of approximately 826,000 markers [18,20]. Imputation to approximately 96 million markers was subsequently carried out using the Haplotype Reference Consortium and UK10K/1000 Genomes reference panels [20]. Clinical biochemistry markers, including total cholesterol, HDL cholesterol, and CRP, were measured at baseline in serum samples. Full details of the biochemistry sampling, handling and quality control protocol, and assay method have been provided previously [21].

**UK Clinical Practice Research Datalink.**   To estimate the potential for disease prevention in a general population setting, we used data from the CPRD, a primary care database of anonymised medical records covering over 11.3 million individuals opting into data linkage from 674 general practices in the UK [22]. Individual-level data from consenting practices in the CPRD have been linked to HES and the national death registry. Details of the CPRD data used and endpoint definition are provided in S2 Text. The present analysis involved records of 2.1 million patients, a random sample of all CPRD data, working under the assumption that individuals in this database should be broadly representative of the UK general population.

## Statistical analysis

To approximate populations relevant to CVD primary prevention, we focused on first-onset CVD outcomes among those with no prior history of CVD and not taking lipid-lowering treatments at recruitment. Analyses were performed according to a pre-specified analysis plan (S1 Analysis Plan) and restricted to participants of self-reported European ancestry, excluding those who (1) had missing genotype array or conventional risk factor information; (2) had a history of CVD at baseline (i.e., CHD, other heart disease, stroke, transient ischaemic attack, peripheral vascular disease, angina, or cardiovascular revascularization); (3) used lipid-lowering treatment at baseline; or (4) were included in the dataset to estimate component score mixing weights during PRS construction (see S1 Fig). The primary outcome was a first-onset CVD

event, defined as the composite of CHD (i.e., myocardial infarction or fatal CHD) or any stroke. Secondary outcomes included each of CHD and stroke separately, and a combination of CHD, stroke, and cardiac revascularisation procedures (i.e., percutaneous transluminal coronary angioplasty [PTCA] and coronary artery bypass grafting [CABG]) (S1 Table).

We used separate PRSs for CHD and ischaemic stroke as 2 independent variables to predict the composite CVD outcome. PRSs were previously constructed using a meta-score approach based on external summary statistics from the previous largest genome-wide association studies (GWASs) of CHD and stroke [23,24]. Detailed information on PRS derivation has been previously provided [16,17], and the PRSs are publicly available and annotated at the PGS Catalog (http://www.pgscatalog.org) under accessions PGS000018 and PGS000039, respectively. The PRS for CHD comprised 1,743,979 variants where the mixing weights of component scores were estimated using 3,000 participants in UKB. The PRS for ischaemic stroke included 2,759,740 variants where the mixing weights of component scores were estimated using 12,000 participants in UKB (including the 3,000 participants mentioned above). Participants used in the training dataset were excluded from subsequent analysis. Previous analyses have not found evidence of overfitting [16,17], and independent replications have demonstrated consistent effect sizes [25–27]. The partial Pearson correlation coefficient between the PRS for CHD and the PRS for stroke was 0.32. In sensitivity analyses we (1) replaced the PRS for ischaemic stroke with a PRS for all stroke and (2) used a single PRS for the composite CVD outcome.

HRs were calculated using Cox proportional hazards models, stratified by UKB recruitment centre and sex, and using time since study entry as the timescale. Outcomes were censored if a participant was lost to follow-up or died from non-CVD causes, or if the end of available follow-up was reached (for England: 31 March 2017; Scotland: 30 October 2016; Wales: 30 May 2016). Predictors were entered as linear terms, after visual checking for log-linearity. No violation of the proportional hazards assumption was identified. Sensitivity analyses included calculation of cumulative incidence of CVD outcomes based on the cause-specific hazards estimated from Cox regression, in the presence of competing risk from non-CVD deaths [28,29].

The incremental predictive ability of PRSs for CHD and stroke was assessed upon addition (as 2 separate linear terms) to a model containing age, sex, systolic blood pressure, smoking status, history of diabetes, and total and HDL cholesterol (i.e., conventional risk factors). Risk discrimination was assessed using Harrell's C-index, stratified by UKB recruitment centre and sex [30]. To avoid overestimation of the model's ability to predict risk, we applied an internal/external validation approach by validating within a subset (i.e., 1 study centre or a 10% randomly selected population in UKB) the prediction model derived from the remaining datasets. Results were then meta-analysed across all validation subsets, weighted by the number of events in that specific subset. Improvements in risk prediction were also quantified by the net reclassification index (NRI), which summarises appropriate directional change in risk predictions for those who do and do not experience an event during follow-up (with increases in predicted risk being appropriate for cases and decreases being appropriate for non-cases) [31,32]. Calibration was assessed by comparing the observed and predicted risks across deciles of predicted risk, and by calculating calibration slope, root mean square error, and the Greenwood–Nam–D'Agostino $p$-value [13,14,33] using a 10-fold cross-validation approach to avoid optimism.

To assess the population health relevance of adding PRSs to conventional risk factors, we generalised our reclassification analyses to the context of a UK population eligible for primary prevention screening (S3 Text). Using CPRD data we recalibrated risk prediction models derived in UKB to give 10-year risks that would be expected in such a UK primary care setting, employing methods previously described [34]. (Since 10 years of follow-up was not available

for all UKB recruitment centres, we used 9-year risk estimates in reclassification analyses.) Details are provided in S3 Text.

We modelled a population of 100,000 adults aged 40–75 years in the CPRD, with an age and sex profile matching that of the contemporary UK population (2017 mid-year population) [35], and CVD incidence rates as observed in individuals without previous CVD and not taking statins. We assumed an initial policy of statin allocation for people at ≥10% predicted 10-year risk as recommended by National Institute for Health and Care Excellence (NICE) guidelines [7]. We then modelled additional targeted assessment of PRSs, or CRP, among people at intermediate risk (5% to <10% predicted 10-year risk) to estimate the potential for additional treatment allocation and case prevention, assuming statin allocation would reduce CVD risk by 20% [36]. Details are in S3 Text. Analyses were performed with PLINK 2.0 [37] and Stata version 14, with 2-sided p-values and 95% confidence intervals. This study follows TRIPOD reporting guidelines (S1 TRIPOD Checklist).

## Results

### Characteristics of the study participants and association with CVD outcomes

Of the 502,219 participants initially enrolled in UKB, 306,654 participants met the inclusion criteria for this analysis: self-reported European ancestry, without a history of CVD, not on lipid-lowering treatment, and with complete information on genotype array data and conventional risk predictors (Table 1). During 2.6 million person-years at risk (median [5th, 95th percentile] follow-up of 8.1 [6.8–9.4] years), 5,680 CVD cases were recorded, including 3,333 CHD and 2,347 stroke events. Fig 2 shows the baseline characteristics of participants, as well as HRs for CVD adjusted for conventional risk factors. HRs for CHD and stroke outcomes separately and for the composite secondary outcome (including CHD, stroke, PTCA, and CABG) are presented in S2 Fig. Both PRSs showed log-linear associations with CVD outcomes, with

**Table 1. Baseline characteristics of UK Biobank participants who had no prior history of vascular disease and were not on lipid-lowering treatment, by sex (n = 306,654).**

| Baseline characteristic | Female | Male | Total |
|---|---|---|---|
| Number of participants | 174,773 | 131,881 | 306,654 |
| Age at recruitment, years | 56.0 (7.9) | 55.9 (8.2) | 56.0 (8.0) |
| **Cardiovascular risk factors** | | | |
| Current-smoker, percent | 9.3 | 11.7 | 10.3 |
| History of diabetes, percent | 0.8 | 1.7 | 1.2 |
| Treatment of hypertension, percent | 10.9 | 11.7 | 11.2 |
| Systolic blood pressure, mm Hg | 134.2 (18.6) | 140.4 (17.3) | 136.9 (19.1) |
| Total cholesterol, mmol/l | 6.0 (1.1) | 5.8 (1.0) | 5.9 (1.1) |
| HDL cholesterol, mmol/l | 1.6 (0.4) | 1.3 (0.3) | 1.5 (0.4) |
| LDL cholesterol, mmol/l | 3.7 (0.8) | 3.7 (0.8) | 3.7 (0.8) |
| C-reactive protein, Ln, mg/l | 0.3 (1.1) | 0.3 (1.0) | 0.3 (1.1) |
| **Incident cardiovascular outcomes** | | | |
| Follow-up, years, median (5th–95th percentile) | 8.2 (6.8–9.4) | 8.1 (6.5–9.3) | 8.1 (6.8–9.4) |
| Number of coronary heart disease cases | 2,453 | 880 | 3,333 |
| Number of stroke cases | 1,311 | 1,036 | 2,347 |

Data are shown as mean (SD), unless otherwise stated, adjusted for UK Biobank study centre.

HDL, high-density lipoprotein; LDL, low-density lipoprotein.

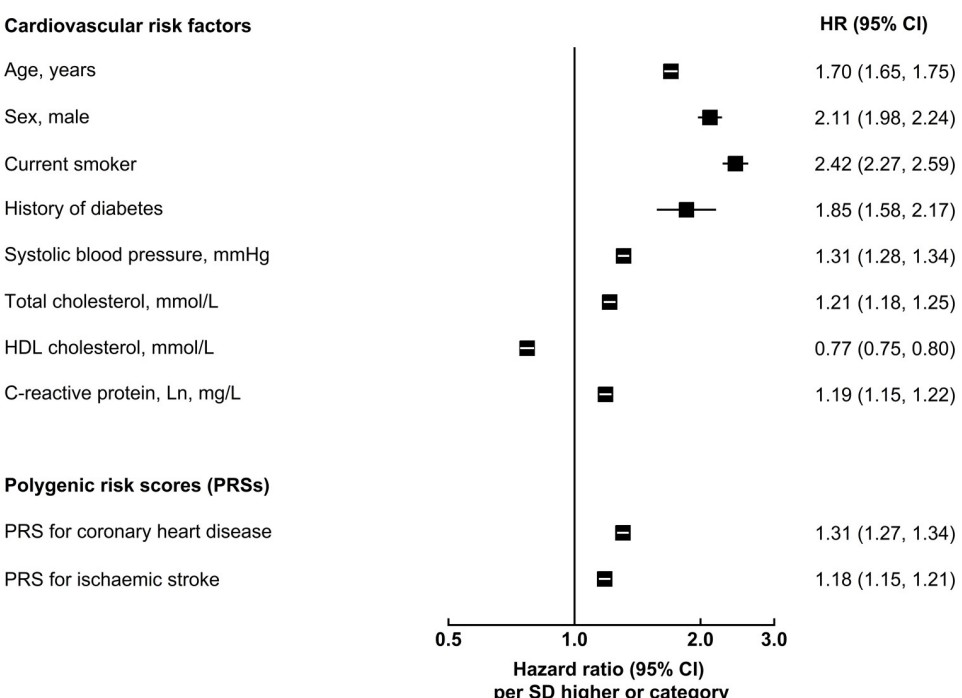

**Fig 2. Adjusted hazard ratios of conventional cardiovascular risk factors and polygenic risk scores for first-onset cardiovascular outcomes.** Hazard ratios (HRs) were estimated using Cox regression, stratified by study centre and sex, and adjusted for age at baseline, smoking status, history of diabetes, systolic blood pressure, total cholesterol, and high-density lipoprotein (HDL) cholesterol levels, where appropriate. For continuous variables, HRs are shown per SD higher of each predictor to facilitate comparison. For categorical variables, HRs are shown for men versus women, for patients with diabetes versus without, and for current smokers versus others.

HRs of 1.57 (95% CI 1.51–1.62) for CHD and 1.19 (95% CI 1.14–1.24) for stroke, after adjustment for age only (S3 Fig). HRs per 1-SD higher PRS did not materially change after adjustment for conventional risk factors; HRs were similar across people with different levels of risk factors, including family history of CVD (S4 and S5 Figs).

## Incremental value in risk prediction

We assessed the incremental predictive ability of PRSs using measures of risk discrimination and reclassification, adding PRSs for CHD and stroke as 2 independent linear terms to a model containing conventional CVD risk factors. For the CVD outcome, the C-index was 0.710 (95% CI 0.703–0.717) for a prediction model containing conventional risk factors alone. The addition of information on PRSs increased the C-index by 0.012 (95% CI 0.009–0.015; Fig 3), yielding a continuous NRI of 10.2% (95% CI 7.2%–13.2%) among CVD cases and 12.6% (95% CI 12.2%–13.0%) among non-cases (Table 2). By comparison, the C-index increased by 0.004 (95% CI 0.003–0.006; Fig 3) after adding information on CRP to the conventional model. The improvement in NRI was also less with addition of CRP than with addition of PRSs, with incident cases more often correctly increased in risk by addition of PRSs (Table 2). Models including PRSs showed good calibration, with good agreement between the observed and predicted CVD risks (S6 Fig).

In hypothesis-generating analyses, the C-index changes with PRSs were possibly somewhat higher in men than women, and in participants with higher total cholesterol, lower HDL

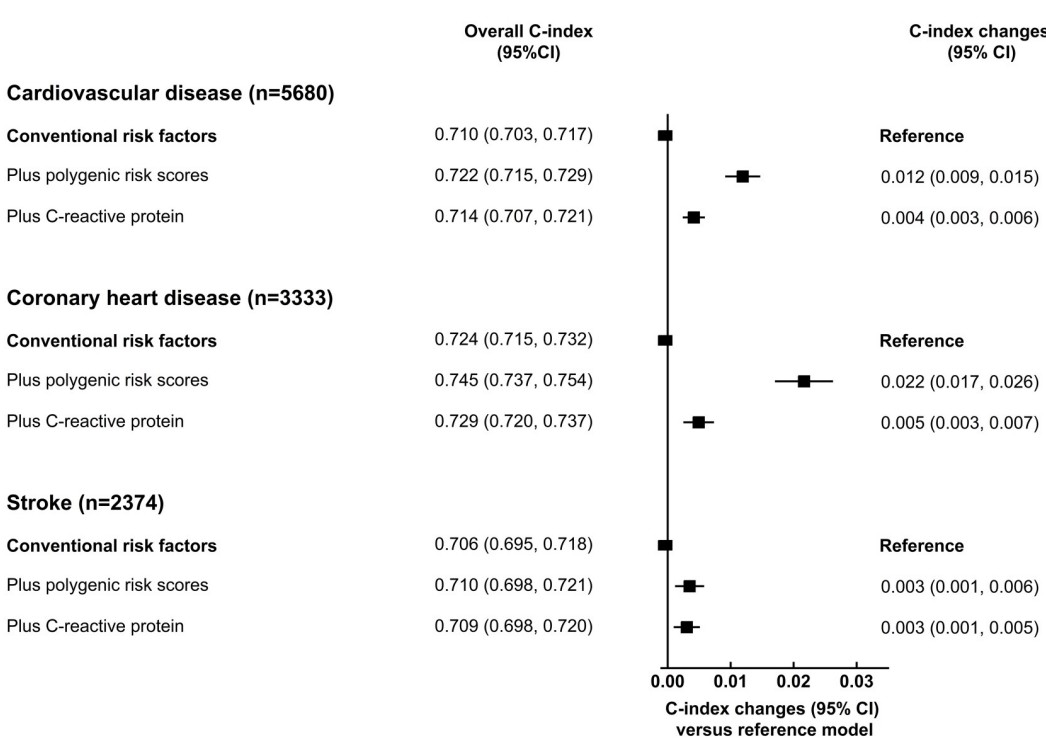

**Fig 3. Incremental predictive ability of polygenic risk scores and C-reactive protein for cardiovascular disease, above conventional risk factors.** Conventional risk factors included age at baseline, sex, smoking status, history of diabetes, systolic blood pressure, total cholesterol, and high-density lipoprotein cholesterol. C-index and related changes were estimated using Cox regression, stratified by study centre and sex, adjusted for age at baseline, smoking status, history of diabetes, systolic blood pressure, total cholesterol, and high-density lipoprotein cholesterol. 95% confidence intervals (CIs) were estimated using the efficient jackknife approach.

cholesterol, and higher predicted 10-year CVD risk (Fig 4; S2 Table). Among CVD cases and controls, continuous NRIs with assessment of PRSs were 11.5% (95% CI 7.8%–15.1%) and 14.1% (95% CI 13.5%–14.6%) in men, and 8.3% (95% CI 3.1%–13.5%) and 8.8% (95% CI 8.3%–9.3%) in women, respectively (S3 Table). The predictive value of PRSs was greater for CHD than for stroke outcomes (Table 2; Fig 3 and S7 Fig).

**Table 2. Net reclassification index (NRI) for cardiovascular disease (generalised to a primary prevention population) with addition of information on polygenic risk scores or C-reactive protein, above conventional risk factors.**

| Factors included | Continuous NRI (95% CI) versus conventional risk factors alone | | |
|---|---|---|---|
| | Cardiovascular disease (*n* = 5,680) | Coronary heart disease (*n* = 3,333) | Stroke (*n* = 2,347) |
| **Conventional risk factors plus polygenic risk scores** | | | |
| Non-cases | 12.6 (12.2, 13.0) | 17.5 (17.1, 17.9) | 6.6 (6.2, 7.0) |
| Cases | 10.2 (7.2, 13.2) | 14.6 (10.8, 18.4) | 3.5 (−1.2, 8.2) |
| **Conventional risk factors plus C-reactive protein** | | | |
| Non-cases | 12.0 (11.6, 12.4) | 12.6 (12.2, 13.0) | 9.9 (9.5, 10.2) |
| Cases | 2.1 (−1.1, 4.9) | 3.8 (0.1, 7.6) | 0.8 (−4.0, 5.5) |

Conventional risk factors included age at baseline, sex, smoking status, history of diabetes, systolic blood pressure, total cholesterol, and high-density lipoprotein cholesterol, with stratification by study centre and sex, where appropriate.

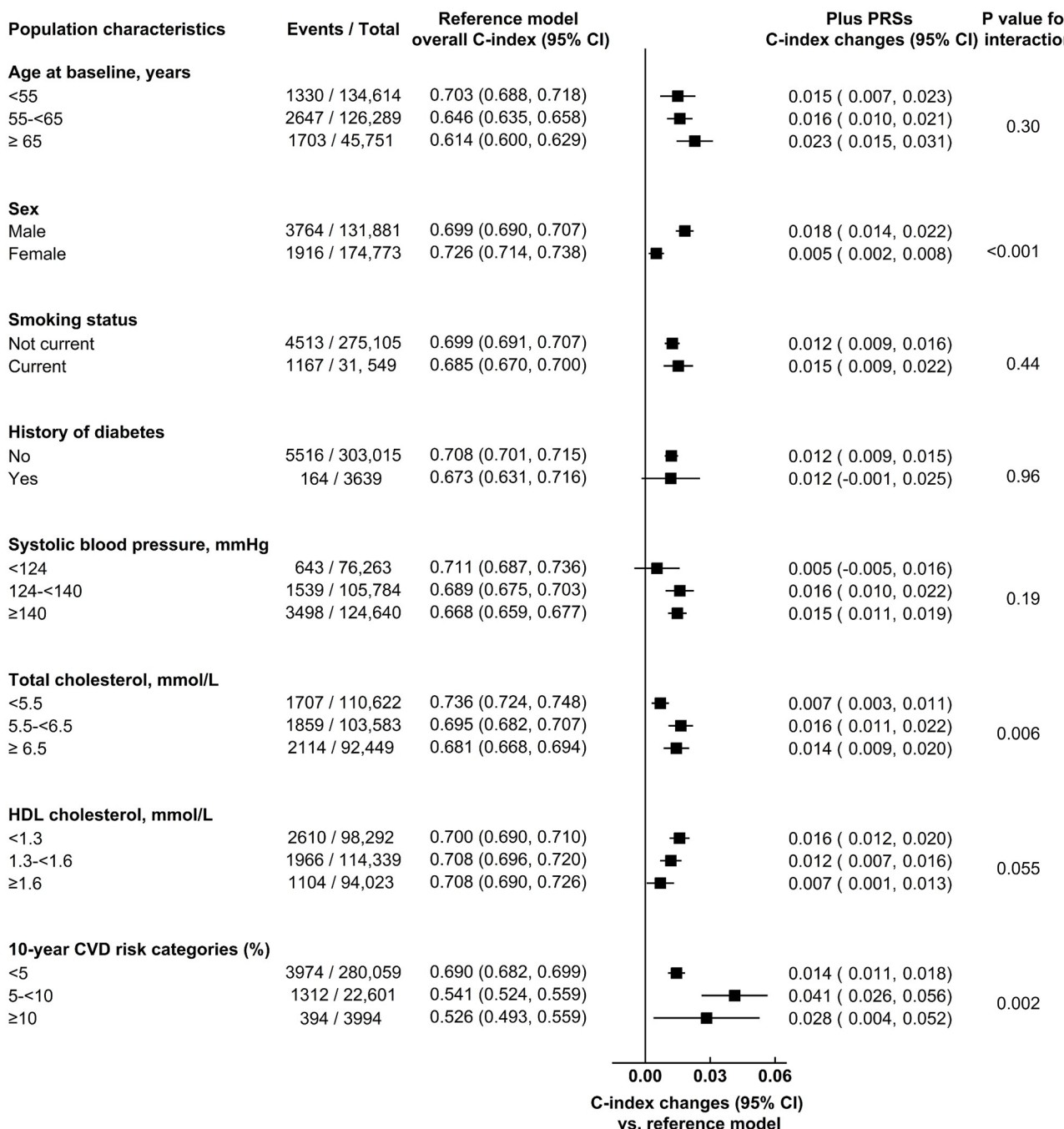

**Fig 4. Incremental predictive ability of polygenic risk scores (PRSs) for cardiovascular disease (CVD) outcomes, beyond conventional risk predictors, across different baseline population characteristics.** The base model included information on the conventional risk factors, i.e., age at baseline, sex, smoking status, history of diabetes, systolic blood pressure, total cholesterol, and high-density lipoprotein (HDL) cholesterol, with stratification by study centre and sex, where appropriate. The prediction model within each subgroup was constructed using coefficients estimated among the entire population.

The results of adding information on PRSs were broadly similar to those observed overall in analyses that included (1) information on body-mass index, family history of CVD, use of blood-pressure-lowering treatment, or CRP in the prediction model (S4 Table; S8 Fig); (2) participants receiving lipid-lowering treatment at baseline (S5 Table; S9 Fig); (3) use of PRSs

derived for the composite CVD outcome or for all stroke (S6 Table); and (4) a broader definition of the CVD outcome (i.e., CHD, stroke, PTCA, or CABG; S9 Fig). Furthermore, similar results were observed in analyses using the internal/external cross-validation approach (S10 and S11 Figs), the Pooled Cohort Equations (S7 Table), or competing risk models for non-CVD deaths (S8 Table).

## Estimate of the potential for disease prevention

In population health modelling, we used age- and sex-specific incidence data from 2.1 million individuals in the CPRD without previous CVD and not taking statins to recalibrate risk models and achieve a predicted risk distribution as would be expected in this primary care population (S3 Text). We translated age- and sex-specific targeted assessment of PRSs to a population of 100,000 adults aged 40–75 years, assuming the age and sex structure of the current UK population, and CVD incidence rates observed in UK primary care. Under this scenario, we estimated that, using conventional risk factors alone, there would be 23,973 individuals classified as having intermediate 10-year (i.e., 5% to <10%) risk who were not already taking or eligible for statin treatment (i.e., people without a history of diabetes or CVD, and with low-density lipoprotein (LDL) cholesterol < 5.0 mmol/l; Fig 5). Additional assessment of PRSs in these individuals (i.e., a 'targeted' approach focusing only in people judged to be at intermediate 10-year risk of CVD after initial screening with conventional risk factors alone) would reclassify 3,115 intermediate-risk individuals as high-risk (i.e., ≥10%), of whom approximately 357 would be expected to have a CVD event within 10 years. This would correspond to an increase of about 7.1% (357/5,054) of the CVD events already classified at high risk using conventional risk predictors alone.

Assuming statin allocation per current guidelines (i.e., those with 10-year CVD risk ≥ 10%) and statin treatment conferring a 20% relative risk reduction, such targeted assessment of PRSs among the intermediate-risk group would help prevent 72 (i.e., $357 \times 0.2$) events over the next 10-year period. In other words, targeted assessment of PRSs in individuals at intermediate risk for a CVD event could help prevent 1 additional event over 10 years for every 336 people so screened. For comparison, the number needed to screen with targeted assessment of CRP would be 491 (S9 Table). Similar results were observed when analysis involved cutoffs for clinical risk categories defined by other guidelines (S10 Table; S12 Fig).

In contrast with the targeted approach, we also modelled a blanket population-wide strategy of additional assessment of PRSs in all adults aged 40–75 years eligible for CVD primary prevention. In this scenario, compared to using conventional risk factors alone, 3,128 individuals would be reclassified from low or intermediate risk (i.e., <10%) to high risk (i.e., ≥10%), and 3,405 individuals would be reclassified from high risk to low or intermediate risk, of whom approximately 358 and 271 would be expected to have a CVD event within 10 years, respectively (S11 Table; S13 Fig), suggesting the need to screen 5,747 people with additional assessment of PRSs to help prevent 1 additional event over 10 years.

## Discussion

We conducted complementary analyses in UKB, a purpose-designed prospective study of about 500,000 individuals, and the CPRD, a cohort of 2.1 million people derived from an extract of contemporary computerised records from general practices in the UK. Overall, our results suggest that the addition of PRSs to conventional risk factors can provide modest improvement in prediction of first-onset CVD, which, if applied at scale, could help prevent 7% more CVD events than use of conventional risk factors alone. Our results have potential

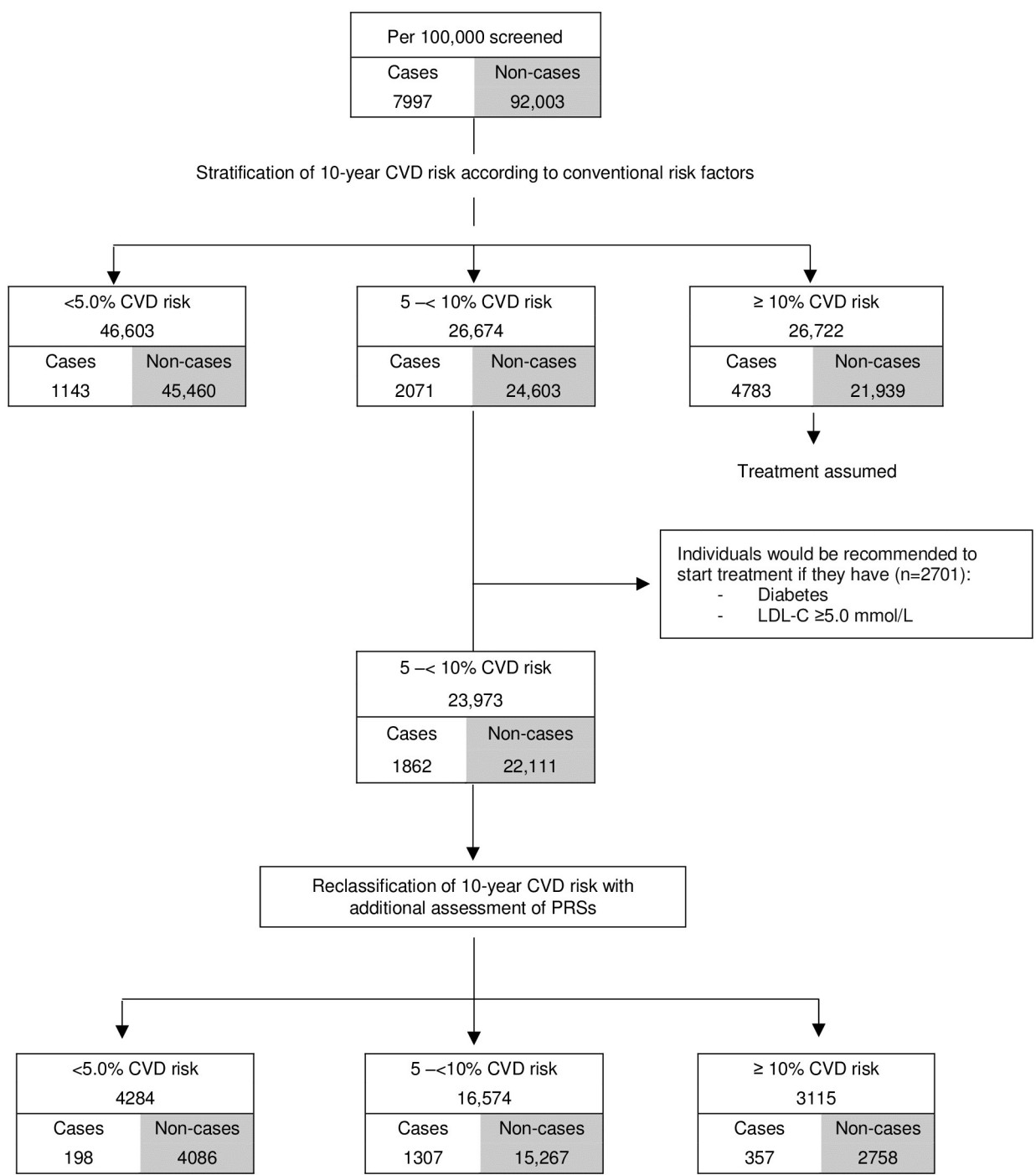

**Fig 5. Estimated public health impact with targeted assessment of polygenic risk scores among 100,000 UK adults in a primary care setting.**
CVD, cardiovascular disease; LDL, low-density lipoprotein; PRS, polygenic risk score.

implications for CVD risk prediction and for the evaluation of the potential population health utility of PRSs for disease.

First, our modelling suggests that, if applied to the contemporary UK population aged 40–75 years [38], additional use of PRSs could help prevent at least several thousand CVD events over the next 10 years beyond assessment of conventional risk factors alone.

Second, our results suggest that targeted use of PRSs would be almost 15 times more efficient than blanket population-wide use. In a modelled scenario in which PRSs were assessed in a primary care setting only among individuals considered at intermediate CVD risk after initial screening with conventional risk predictors alone, we estimated that such targeted assessment of PRSs could reclassify approximately 12% of screened individuals to the high-risk category, of whom 11% would be expected to have a CVD event within 10 years. If such a targeted approach were to be coupled with initiation of statin therapy in accordance with guidelines, our data suggest 1 extra CVD outcome could be prevented over a period of 10 years for approximately every 340 people in whom PRSs are assessed (compared with the need to screen approximately 5,700 people to achieve the same gain when using a blanket screening approach).

Third, as a benchmark, we compared the incremental predictive gains afforded by assessment of PRSs with those provided by CRP measurement (a plasma biomarker recommended for screening in some primary prevention guidelines) [12,15], with our results demonstrating a >1.5-fold greater gain in predictive accuracy with PRSs than CRP.

Fourth, we found that assessment of PRSs could improve prediction of CHD much more than prediction of stroke. Further work is needed to understand fully the reasons for such differential gains, which may relate both to the greater phenotypic heterogeneity of stroke outcomes [39–41] and the relatively lower statistical power of previous GWASs of stroke [24,41] compared with CHD [23]. It is likely that the composite outcome of CVD involves greater phenotypic and genetic heterogeneity than either CHD or stroke alone. Nevertheless, our study used the primary outcome of any first CVD event (defined as fatal or nonfatal CHD or stroke), in keeping with current CVD primary prevention guidelines that promote joint prediction and prevention of CHD and stroke.

Our study had major strengths. In the analysis of UKB, we approximated the targeted population for CVD primary prevention efforts by focusing on >300,000 participants without a history of CVD at baseline who were not taking lipid-lowering treatment. For these participants, we had access to concomitant and nearly complete information on several conventional CVD risk factors (e.g., lipid measurements) as well as on PRSs. We used multiple complementary metrics of risk discrimination and reclassification, as well as different absolute risk thresholds used in different clinical guidelines. The broadly concordant results we observed across these metrics supported the validity of our main conclusions. To extend the relevance of our findings to a UK primary care population, we also conducted modelling using the UK CPRD, adapting (recalibrating) our findings from UKB to be more representative of the general population. This adaptation was important because the general UK population has a higher baseline risk for CVD than the volunteers who enrolled in UKB, underscoring the need for recalibration when using established risk thresholds, and before making judgements about the population health utility of PRSs.

Our study also had limitations. We studied only middle-aged European ancestry participants in the UK, which limits the generalisability of our results. Hence, we (and others) are now addressing this gap by conducting studies of PRSs for CVD in different ethnic groups, as well as in other countries. Our study also lacked a health economic evaluation, which was beyond the scope of present analysis. We acknowledge the importance of health economic evaluations as part of future considerations to assess the clinical utility of PRSs for CVD prevention, noting that genome-wide array genotyping has a one-time cost (approximately £25 at current prices in the UK) and can be used to calculate PRSs for CVD as well as for many other chronic diseases. In particular, future studies (including health economic evaluations) are needed to evaluate a range of different CVD screening strategies, including a 'genome first' approach that inverts the current 'conventional risk factors first' approach to CVDs.

Our study did not assess potential psychological harms of using genetic information in CVD risk prediction. However, a previous randomised trial has excluded material effects of this type [42]. We used a conventional 10-year timeframe and standard clinical risk categories, acknowledging that reclassification analyses are intrinsically sensitive to choices of follow-up interval and clinical risk categories. Although we used 9-year risk estimates in reclassification analyses because 10 years of follow-up was not available for all UKB recruitment centres, it had minimal influence on our results. Somewhat greater population health impact than suggested by our analysis would be estimated if we had used less conservative modelling assumptions (e.g., more effective statin regimens, longer time horizons), conventional risk factor weights that were not fitted to UKB, or alternative disease outcomes (e.g., an exclusive focus on CHD). Conversely, our models could have overestimated the potential benefits of assessing PRSs because not all people eligible for statins will receive them or be willing and able to take them and adherent.

In conclusion, our results suggest that the addition of PRSs to conventional risk factors can modestly enhance the prediction of first-onset CVD and could translate into population health benefits if used at scale.

## Supporting information

**S1 Analysis Plan.**
(DOCX)

**S1 Fig. Exclusion criteria applied in derivation of the primary analytic dataset.** *Prior history of cardiovascular disease at baseline included coronary heart disease, angina, other heart disease, stroke, transient ischaemic attack, peripheral vascular disease, and cardiac revascularisations.
(TIF)

**S2 Fig. Hazard ratios for coronary heart disease, stroke, and the composite cardiovascular disease outcome (including coronary heart disease, stroke, and cardiac revascularisations), adjusted for conventional risk factors.** Hazard ratios (HRs) were estimated using Cox regression, stratified by study centre and sex, and adjusted for age at baseline, smoking status, history of diabetes, systolic blood pressure, total cholesterol, and HDL cholesterol, where appropriate. For continuous variables, HRs are shown for each SD higher of each predictor to facilitate comparison. For categorical variables, HRs are shown for men versus women, for patients with diabetes versus without, and for current smokers versus others.
(TIFF)

**S3 Fig. Shape and strength of associations of polygenic risk scores with risk of coronary heart disease and stroke.** The shape of association was estimated by dividing all participants into fifths. Hazard ratios were estimated using Cox regression, stratified by study centre and sex, and adjusted for age at baseline. Each square has an area inversely proportional to the effective variance of the log risk in that specific group, with vertical lines representing the 95% confidence intervals.
(TIFF)

**S4 Fig. Hazard ratios of polygenic risk scores (PRSs) for coronary heart disease and stroke, after progressive adjustment for conventional cardiovascular risk factors.** Hazard ratios were estimated using Cox regression, stratified by study centre and sex, and adjusted for conventional cardiovascular risk factors, where appropriate.
(TIFF)

**S5 Fig. Adjusted hazard ratios of polygenic risk scores for incident coronary heart disease and stroke by population characteristics at baseline.** Hazard ratios were estimated using Cox regression, stratified by study centre and sex, and adjusted for conventional cardiovascular risk factors, where appropriate.
(TIFF)

**S6 Fig. Observed and predicted cardiovascular risk when adding information on polygenic risk scores and/or C-reactive protein to conventional risk factors, in UK Biobank.** PRS, polygenic risk score; CRP, C-reactive protein; RMSE, root mean square error; GND Chi-sq, Greenwood–Nam–D'Agostino chi-squared index. Conventional risk factors included age at baseline, sex, smoking status, history of diabetes, systolic blood pressure, total cholesterol, and HDL cholesterol levels. Polygenic risk scores included the polygenic risk score for coronary heart disease and the one for ischaemic stroke (see Fig 2) as 2 linear predictors in the model throughout. To avoid optimism in assessing calibration of the models, we applied a 10-fold cross-validation approach. We divided the datasets into 10 random subsets with the same number of participants, with prediction models developed within 90% of the dataset, and validated using the remaining 10% of the dataset. Each blue square in the plots represents the mean value of the predicted/observed risk within each decile; these values were pooled across the 10 validation subsets and weighted by the number of events in that group. The ratios were calculated as the ratio of observed risks to predicted risks, with 1 representing perfect calibration. The RMSE was used to assess the differences between the predicted risks and the observed risks. The Greenwood–Nam–D'Agostino test is an extension of the Hosmer–Lemeshow test to situations with censored survival data, and tests the null hypothesis that the observed and expected probabilities are identical in each group.
(TIFF)

**S7 Fig. Incremental predictive ability of polygenic risk scores (PRSs) for cardiovascular disease, above conventional risk factors.** Conventional risk factors included age at baseline, sex, smoking status, history of diabetes, and systolic blood pressure, with or without baseline measurements of total cholesterol and HDL cholesterol.
(TIFF)

**S8 Fig. Incremental predictive values of polygenic risk scores (PRSs), above conventional risk factors, including body-mass index or family history of cardiovascular disease in the reference model.** CVD, cardiovascular disease. Conventional risk factors included age at baseline, sex, smoking status, history of diabetes, systolic blood pressure, total cholesterol, and HDL cholesterol. Polygenic risk scores included the polygenic risk score for CHD and the one for ischaemic stroke (see Fig 2) as 2 linear predictors in the model throughout.
(TIFF)

**S9 Fig. Incremental predictive values of polygenic risk scores (PRSs) above conventional risk factors, among individuals with or without lipid-lowering treatment, and for different cardiovascular outcomes.** Conventional risk factors included age at baseline, sex, smoking status, history of diabetes, systolic blood pressure, total cholesterol, and HDL cholesterol. Polygenic risk scores included the polygenic risk score for coronary heart disease and the one for ischaemic stroke (see Fig 2) as 2 linear predictors in the model throughout. Cardiac procedures included cardiovascular outcomes identified via OPCS-4: K40–K46, K49, K50.1, K50.2, K50.4, or K75.
(TIFF)

**S10 Fig. Incremental predictive value of polygenic risk scores, above conventional risk factors, by 10-fold cross-validation in UK Biobank.** Polygenic risk scores included the polygenic risk score for CHD and the one for ischaemic stroke (see Fig 2) as 2 linear predictors in the model throughout. Each subset represented a 10% randomly selected subset of UK Biobank participants from the entire study population. The prediction model was derived using 90% of the dataset, and validated among the remaining 10%. The overall C-index and relevant changes were estimated by meta-analysing the subset-specific results, weighted by the number of events in that subset.
(TIFF)

**S11 Fig. Incremental predictive value of polygenic risk scores, above conventional risk factors, when leaving 1 recruitment centre out per iteration in UK Biobank.**
(TIFF)

**S12 Fig. Estimates of public health impact with targeted assessment of polygenic risk scores, among 100,000 UK adults using American Heart Association/American College of Cardiology guideline.**
(DOCX)

**S13 Fig. Estimates of public health impact of additional assessment of polygenic risk scores or C-reactive protein, above conventional risk factors, among 100,000 individuals.** Numbers in red are shown for individuals who were initially classified as being at high risk and were reclassified down to intermediate risk; numbers in blue are shown for individuals moving from intermediate risk to high risk. Among 100,000 individuals, 1,197 cases and 7,354 non-cases were treated, irrespective of their 10-year CVD risk, since they had history of diabetes or LDL cholesterol $\geq$ 5.0 mmol/l. The number of cases screened as high risk, or classified as high risk due to diabetes or LDL cholesterol level, using conventional risk factors alone was 5,054, and thus, the number of events prevented was 1,011 (5,054 $\times$ 0.2).
(TIF)

**S1 Table. Definition of study outcomes.**
(DOCX)

**S2 Table. Incremental predictive ability of polygenic risk scores, and C-reactive protein, above conventional risk factors, by population characteristics at baseline.** Conventional risk factors included age at baseline, sex, smoking status, history of diabetes, systolic blood pressure, total cholesterol, and HDL cholesterol. Prediction model was developed using Cox regression for all participants, stratified by study centre and sex, adjusted for conventional risk predictors, where appropriate. Polygenic risk scores included the polygenic risk score for CHD and the one for ischaemic stroke (see Fig 2) as 2 linear predictors in the model throughout.
(DOCX)

**S3 Table. Net reclassification index (NRI) for incident cardiovascular disease by addition of information on polygenic risk scores, and C-reactive protein, above conventional risk factors, for non-cases and cases.** ACC, American College of Cardiology; AHA, American Heart Association; NICE, National Institute for Health and Care Excellence. *Conventional risk factors included age, sex, smoking, systolic blood pressure, history of diabetes, total cholesterol, and HDL cholesterol. Polygenic risk scores included the polygenic risk score for CHD and the one for ischaemic stroke (see Fig 2) as 2 linear predictors in the model throughout. Calculations of the above categorical NRIs were <5%, 5% to <7.5%, and $\geq$7.5% according to the 2019 ACC/AHA guideline, and <5%, 5% to <10%, and $\geq$10% according to the 2014 NICE

guideline.
(DOCX)

**S4 Table. Partial likelihood ratio test for models with polygenic risk scores beyond conventional risk factors, C-reactive protein, and treatment of hypertension.** Conventional risk factors included age at baseline, sex, smoking status, history of diabetes, systolic blood pressure, total cholesterol, and HDL cholesterol, with stratification by study centre and sex, where appropriate.
(DOCX)

**S5 Table. Net reclassification index (NRI) for incident cardiovascular disease by addition of information on polygenic risk scores, and C-reactive protein, above conventional risk factors, for non-cases and cases, including participants on statin treatment at baseline.** ACC, American College of Cardiology; AHA, American Heart Association; NICE, National Institute for Health and Care Excellence. *Conventional risk factors included age, sex, smoking, systolic blood pressure, history of diabetes, total cholesterol, and HDL cholesterol. Polygenic risk scores included the polygenic risk score for CHD and the one for ischaemic stroke (see Fig 2) as 2 linear predictors in the model throughout. Calculations of the above categorical NRIs were <5%, 5% to <7.5%, and ≥7.5% according to the 2019 ACC/AHA guideline, and <5%, 5% to <10%, and ≥10% according to the 2014 NICE guideline.
(DOCX)

**S6 Table. Comparison of different polygenic risk scores (PRSs) on strength of associations, discriminative ability, and reclassification index for different cardiovascular outcomes, in UK Biobank.** CVD, cardiovascular disease; CHD, coronary heart disease; IS, ischaemic stroke; CI, confidence interval. Conventional risk factors included age at baseline, sex, smoking status, history of diabetes, systolic blood pressure, total cholesterol, and HDL cholesterol, with stratification by study centre and sex, where appropriate. The PRS for CHD and the PRS for IS were constructed using methods as in our previous work [1]. The PRS for stroke was constructed using the genome-wide significant variants in the MEGASTROKE consortium for total stroke, and linkage-disequilibrium-thinned in UK Biobank, with corresponding weights taken from the MEGASTROKE consortium [2]. Construction procedures of the 2 above PRSs did not include estimates from previous GWASs on other vascular risk factors. The PRS for IS was constructed using methods described in our previous work [3], by taking account of 19 phenotypes, and is publicly available (https://www.pgscatalog.org/score/PGS000039/). The PRS for CVD was constructed using the same approach as the PRS for IS but with CVD as the outcome.
(DOCX)

**S7 Table. Incremental predictive ability of polygenic risk scores, and C-reactive protein, above the updated Pooled Cohort Equations (PCE) score.** *The PCE score for study participants in UK Biobank was calculated using the updated Pooled Cohort Equations score, i.e., the weights for each constituent predictor variable, as previously published [1]. **PCE variables included age at baseline, sex, smoking status, history of diabetes, systolic blood pressure, total cholesterol, HDL cholesterol, ethnicity, and treatment of high blood pressure, weighted by the Cox regression coefficients estimated in UK Biobank. Polygenic risk scores included the polygenic risk score for CHD and the one for ischaemic stroke (see Fig 2) as 2 linear predictors in the model throughout.
(DOCX)

**S8 Table. Incremental predictive ability of polygenic risk scores, and C-reactive protein, with or without adjusting for competing risk from non-cardiovascular death.** Conventional

risk factors included age at baseline, sex, smoking status, history of diabetes, systolic blood pressure, total cholesterol, and HDL cholesterol. Polygenic risk scores included the polygenic risk score for CHD and the one for ischaemic stroke (see Fig 2) as 2 linear predictors in the model throughout. Cumulative incidence of the composite CVD outcomes was estimated using the cause-specific hazards ratios from Cox regression, in the presence of competing risk from non-CVD deaths.
(DOCX)

**S9 Table. Estimates of public health impact with targeted assessment (intermediate risk: 5% to <10%) of polygenic risk scores, and C-reactive protein, among 100,000 UK adults.** PRS, polygenic risk score; CRP, C-reactive protein. Conventional risk factors included age at baseline, sex, smoking, systolic blood pressure, history of diabetes, total cholesterol, and HDL cholesterol. Polygenic risk scores included the polygenic risk score for CHD and the one for ischaemic stroke (see Fig 2) as 2 linear predictors in the model throughout. The predicted 10-year cardiovascular risk categories used the 2014 NICE guideline. Estimates of public health impact for a hypothetical population of 100,000 individuals (40–75 years) were based on (1) the sex- and age-specific (5-year) profile of a standard UK population (2017 mid-year population) [35] and (2) sex-specific 5-year age-at-risk incidence rates of cardiovascular disease in the CPRD, among individuals without prior history of cardiovascular disease and not on statin treatment at baseline. Estimates for public health impact are shown before and after recalibration.
(DOCX)

**S10 Table. Estimates of public health impact with targeted assessment (intermediate risk: 5% to <7.5%) of polygenic risk scores (PRSs), and C-reactive protein, among 100,000 UK adults.** CRP, C-reactive protein. Conventional risk factors included age at baseline, sex, smoking, systolic blood pressure, history of diabetes, total cholesterol, and HDL cholesterol. Polygenic risk scores included the polygenic risk score for CHD and the one for ischaemic stroke (see Fig 2) as 2 linear predictors in the model throughout. The predicted 10-year cardiovascular risk categories used the 2019 American Heart Association/American College of Cardiology guideline. Estimates of public health impact for a hypothetical population of 100,000 individuals (40–75 years) were based on (1) the sex- and age-specific (5-year) profile of a standard UK population (2017 mid-year population) [35] and (2) sex-specific 5-year age-at-risk incidence rates of cardiovascular disease in the CPRD, among individuals without prior history of cardiovascular disease and not on statin treatment at baseline. Estimates for public health impact are shown before and after recalibration.
(DOCX)

**S11 Table. Numerical results for estimates of public health impact by additional assessment of polygenic risk scores (PRSs) or C-reactive protein, above conventional risk predictors, among 100,000 individuals.** *Among cases and non-cases, respectively, 1,197 and 7,354 participants had diabetes or LDL cholesterol measurement of 5.0 mmol/l or greater. Numbers in red are shown for individuals who were reclassified downwards with additional assessment, and numbers in blue are shown for individuals who were reclassified upwards with additional assessment. Polygenic risk scores included the polygenic risk score for CHD and the one for ischaemic stroke (see Fig 2) as 2 linear predictors in the model throughout.
(DOCX)

**S1 Text. Description of analytic dataset from UK Biobank.**
(DOCX)

**S2 Text. Description of analytic dataset from Clinical Practice Research Datalink.**
(DOCX)

**S3 Text. Statistical methods used for estimating public health impact.**
(DOCX)

**S1 TRIPOD Checklist.**
(DOCX)

## Acknowledgments

The views expressed are those of the authors and not necessarily those of the NHS, the NIHR, or the Department of Health and Social Care.

## Author Contributions

**Conceptualization:** Lisa Pennells, Christopher P. Nelson, Frank Dudbridge, John R. Thompson, Adam S. Butterworth, Angela Wood, John Danesh, Nilesh J. Samani, Michael Inouye, Emanuele Di Angelantonio.

**Data curation:** Scott C. Ritchie, Gad Abraham, Matthew Arnold, Thomas Bolton, Qi Guo, Eleni Sofianopoulou, David Stevens.

**Formal analysis:** Luanluan Sun, Lisa Pennells, Stephen Kaptoge, Christopher P. Nelson, Scott C. Ritchie, Gad Abraham, Matthew Arnold, Steven Bell, Stephen Burgess, Eleni Sofianopoulou, David Stevens, Adam S. Butterworth, Angela Wood, Michael Inouye.

**Funding acquisition:** John Danesh, Nilesh J. Samani, Emanuele Di Angelantonio.

**Methodology:** Lisa Pennells, Stephen Kaptoge, Scott C. Ritchie, Matthew Arnold, Stephen Burgess, Frank Dudbridge, Qi Guo, John R. Thompson, Adam S. Butterworth, Angela Wood, Michael Inouye, Emanuele Di Angelantonio.

**Resources:** John Danesh, Nilesh J. Samani, Emanuele Di Angelantonio.

**Supervision:** Stephen Kaptoge, Adam S. Butterworth, John Danesh, Nilesh J. Samani, Michael Inouye, Emanuele Di Angelantonio.

**Visualization:** Luanluan Sun.

**Writing – original draft:** Luanluan Sun, Lisa Pennells, John Danesh, Nilesh J. Samani, Michael Inouye, Emanuele Di Angelantonio.

**Writing – review & editing:** Lisa Pennells, Stephen Kaptoge, Christopher P. Nelson, Scott C. Ritchie, Gad Abraham, Matthew Arnold, Steven Bell, Thomas Bolton, Stephen Burgess, Frank Dudbridge, Eleni Sofianopoulou, John R. Thompson, Adam S. Butterworth, Angela Wood, John Danesh, Nilesh J. Samani, Michael Inouye, Emanuele Di Angelantonio.

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
