## [Editor Report · Decision Letter 0]

3 Feb 2020

Dear Dr Di Angelantonio, 

Thank you for submitting your manuscript entitled "Adding polygenic risk scores to conventional risk factors 

in cardiovascular disease prediction" for consideration by PLOS Medicine.

Your manuscript has now been evaluated by the PLOS Medicine editorial staff and I am writing to let you know that we would like to send your submission out for external peer review.

Kind regards,

Helen Howard, for Clare Stone PhD 

Acting Editor-in-Chief

PLOS Medicine 

plosmedicine.org

---

## [Decision Letter · Decision Letter 1]

6 Mar 2020

Dear Dr. Di Angelantonio,

Thank you very much for submitting your manuscript "Adding polygenic risk scores to conventional risk factors 

in cardiovascular disease prediction" (PMEDICINE-D-20-00284R1) for consideration at PLOS Medicine. 

[LINK]

In light of these reviews, I am afraid that we will not be able to accept the manuscript for publication in the journal in its current form, but we would like to consider a revised version that addresses the reviewers' and editors' comments. Obviously we cannot make any decision about publication until we have seen the revised manuscript and your response, and we plan to seek re-review by one or more of the reviewers. 

We expect to receive your revised manuscript by Mar 27 2020 11:59PM. Please email us (plosmedicine@plos.org) if you have any questions or concerns.

We look forward to receiving your revised manuscript. 

Sincerely,

Adya Misra, PhD

Senior Editor 

PLOS Medicine

plosmedicine.org

Title- Please revise your title according to PLOS Medicine's style. Your title must be nondeclarative and not a question. It should begin with main concept if possible. "Effect of" should be used only if causality can be inferred, i.e., for an RCT. Please place the study design ("A randomized controlled trial," "A retrospective study," "A modelling study," etc.) in the subtitle (ie, after a colon).

Author summary-At this stage, we ask that you include a short, non-technical Author Summary of your research to make findings accessible to a wide audience that includes both scientists and non-scientists. The Author Summary should immediately follow the Abstract in your revised manuscript. This text is subject to editorial change and should be distinct from the scientific abstract. Please see our author guidelines for more information: https://journals.plos.org/plosmedicine/s/revising-your-manuscript#loc-author-summary

Abstract methods and findings- the last sentence should be a limitation of your study design

Abstract background- please clearly state the aim of your study

Abstract conclusions- please tone down in view of the “modest gains” in prediction accuracy

Data availability- please provide details of how the data may be accessed, providing contact details, URLs or accession numbers as needed

Page 3- please include a header for “Introduction”

Reference square brackets must be placed before a full stop for example [2,5]. 

Methods- please introduce CHD on first view

A table reference is missing on Page 9line 6

Please complete the TRIPOD checklist, and include the completed checklist as Supporting Information. When completing the checklist, please use section and paragraph numbers, rather than page numbers. Please add the following statement, or similar, to the Methods: "This study is reported as per xxx guideline (S1 Checklist)."

Overall the language needs to be toned down as this is an observational study 

Comments from the reviewers:

Reviewer #1: This study investigated the added values of polygenic risk scores in addition to conventional risk factors in prediction of cardiovascular disease. However, there are quite a few major issues needing attention.

1) Cox models were used throughout the study to predict CVD events. As the outcomes here were CVD events other than all cause mortality, there is a major issue of competing risk, which unfortunately was not addressed in the paper.

2) The authors developed the PRSs through 10-fold cross validation using the UK biobank data. However, there was no external validation (using independent external datasets) for the models, which is crucial for the validation of the developed models. Also, there is no calibration of the developed models. Could authors please read and follow the TRIPOD statement?

3) The prediction model containing conventional risk factors alone gave a C-index of 71% while the PRSs increased the C-index by 1% when adding to the model. Although this 1% is statistically significant, it is mainly because the huge sample size which will detect any tiny differences. Firstly this 1% increase hasn't gone through rigorous external validation, secondly the predictive model is not really convincing as hasn't adjusted for competing risk, and finally whole exercise of adding PRSs to the prediction model to increase the precision from 71% to 72% doesn't offer real practical benefits in clinical settings. It's more like a exploratory or association study to show the PRSs are of predictive value in predicting CVD outcomes, however this is already known.

4) The whole excise of using CPRD data to show the benefits of the added value of PRSs is very difficult to follow and not convincing at all, because everything was based on assumptions especially for different projected outcomes. The argument should be based on the solid outcomes not some arbitrary and hypothetical scenarios. 

5) Figure 1 and 2 are a bit strange to put baseline characteristics and odd ratios together. It would be good to be straightforward and clear to have a baseline table to put UK biobank and CPRD data side by side, then have other tables for prediction models, validation, NPI, and etc.

6) A bit confused and unclear on which models the authors worked on, predict stroke? predict CHD? but the CVD events were defined as stroke or CHD? Also, need to describe exactly which are the derivation and validation datasets, and any independent external validation datasets?

Reviewer #2: Sun et al. study the performance of two previously developed and validated polygenic risk scores for coronary heart disease and stroke. The authors assess the improvement in risk prediction on top of conventional risk factors using around 300,000 UK Biobank participants and estimate the clinical benefits by modeling the impact on data from a primary care setting. 

The paper is well written and in line with several recent studies. The added value of this work comes from the recalibration of risk and modeling to a primary care setting in the UK. 

As the authors are projecting the improvements into general benefits in the primary care setting, it is important that the performance of the scores is additionally assessed in other ancestries and not restricted to individuals of European ancestry. 

The stroke PRS was derived from GWAS summary statistics of 19 phenotypes including stroke and coronary artery disease. What is the correlation between the stroke PRS and coronary heart disease PRS? In previous work the authors have combined polygenic risk scores in a metaGRS, why haven't they followed similar strategy here and created such score for CVD?

Both scores were derived and validated in the UK Biobank which raises some questions regarding external validity. This should be discussed. How would the score perform in a non-UK Biobank dataset?

Use of antihypertensive medication is usually included with conventional risk factors and risk equations. Why is it not included here? 

The authors use CRP for comparison with the PRS. How does the PRS perform on top of conventional risk factors that additionally include CRP? 

Reviewer #3: In this manuscript, Sun and colleagues investigate (a) whether the addition of polygenic risk scores for cardiovascular disease to a clinical model yields a change in risk classification at a population level; and (b) whether using this modified score to influence statin prescriptions is likely to yield a reduction in incident cardiovascular disease in a primary prevention population. These questions are timely and important in the field, coming at a moment when the PRS literature is in transition from demonstrating disease association to showing potential public health impact from implementation. 

A substantial body of prior work, with important contributions from these authors, has demonstrated that polygenic risk scores can function as independent risk factors for incident cardiovascular disease (Inouye, M. et al. Genomic Risk Prediction of Coronary Artery Disease in 480,000 Adults: Implications for Primary Prevention. Journal of the American College of Cardiology 2018) and can improve 10-year risk classification when compared to clinical risk factors alone in various cohorts, including similar work in the UK Biobank that was published after this manuscript's submission (Abraham, G. et al. Genomic prediction of coronary heart disease. Eur Heart J 2016; Elliott, J. et al. Predictive Accuracy of a Polygenic Risk Score-Enhanced Prediction Model vs a Clinical Risk Score for Coronary Artery Disease. JAMA 2020). Although the primary dataset and some of the content is similar to the now-published Elliott et al manuscript, this manuscript extends prior work by asking how many incident CVD events might be prevented in the general UK population if statin prescribing patterns were changed by using a model that adds a polygenic score to classical risk factors. Because the costs (such as those due to genotyping and the increased number of statin prescriptions) were not addressed, some important questions raised by this manuscript are left unanswered. Some amount of the novelty was eclipsed by Elliott et al. However, the authors make key novel contributions (by assessing the potential impact of genomic screening to influence preventive statin prescriptions in a general UK population). Nevertheless, a comprehensive assessment would look not only at the potential benefits as described in this manuscript, but also at the cost and potential harms.

Additional comments:

1) The description of the polygenic scores for CAD and stroke was insufficient to convey a complete understanding of how the scores were derived. The main text identified the source studies for the SNP weights (CARDIoGRAMplusC4D and MEGASTROKE, respectively) and indicated that plink 2.0 was used for genetic analysis. Supplementary Figure 4 indicated that "LD thinning" was performed in UK Biobank. "LD thinning" is not a named function in plink, and colloquially it can represent either (a) LD pruning on MAF or (b) LD clumping on GWAS P-value, both of which use r2 cutoffs and could therefore be the indicated method. Several other aspects of polygenic score creation in this manuscript were not clearly answered in the manuscript: most importantly, why were new scores created when polygenic scores using the same GWAS summary statistics have already been developed, tuned in a UK population, and publicly released (both by authors of this manuscript and others)? How was the r2 cutoff chosen and were other r2 cutoffs considered? The Data Availability section addresses UK Biobank and CPRD availability, but not the availability of the subsetted GWAS summary statistics necessary to compute the same scores in other populations; will those be made available? A more comprehensive treatment of the polygenic score will be important for permitting a complete evaluation, and for reproducibility. Alternatively, using polygenic scores that have already been derived and validated, such as scores previously published by coauthors of this manuscript, would sidestep this concern.

2) The authors compare a CVD risk model whose inputs are clinical risk factors (the same as those that are used in the Pooled Cohorts Equation [PCE]) with a model that includes those clinical risk factors plus polygenic scores. It would be useful to also compare the "PCE+polygenic scores" model to the original PCE that uses the original risk factor weights (i.e., the PCE that is actually in clinical use). Doing so would permit a comparison with typical clinical practice, while the current comparison assesses a hypothetical reweighting of the PCE (which may be more optimally calibrated for the studied population, but which is not actually in clinical use) compared to a PCE+polygenic scores model.

3) The authors exclude individuals who are already taking statins from their main analysis, but importantly they provide a sensitivity analysis that includes those individuals. Those without pre-existing disease but who are taking statins for primary prevention form an important subgroup of individuals whose treatment decisions might be impacted by a PRS-informed cardiovascular disease risk score. Their exclusion can influence the apparent effectiveness of risk scores. For example, in the evaluation of the 2013 ACC Pooled Cohorts Equation, a sensitivity analysis that excluded the ~15% of MESA participants on statins substantially improved the C-statistic (by 0.0135 on average), showing that excluding statin users from risk calculators can have an important influence on model performance (see Goff et al, "2013 ACC/AHA Guideline on the Assessment of Cardiovascular Risk," 2014, Circulation, supplement pages 41-42). Here, the concern would be that excluding current statin-takers might differentially affect a risk score with a polygenic score compared to a risk score without a polygenic score. The authors' sensitivity analysis addresses that concern. It would be helpful to provide more comprehensive results from this sensitivity analysis (e.g., sensitivity, specificity, reclassification counts) rather than just showing the AUC change in Supplementary Figure 9.

4) The likelihood ratio test (LRT) is the uniformly most powerful test comparing models with and without additional predictors. As such, while I believe that the authors have already chosen reasonable statistics to demonstrate clinical impact, for comprehensiveness it could be helpful to include a likelihood ratio test result to more precisely define the statistical improvement of adding the PRS (i.e., LRT of baseline vs baseline+PRS). In addition, a likelihood ratio test would permit a more concise answer to the question of whether the PRS adds information beyond C-reactive protein (i.e., LRT of baseline+CRP vs baseline+CRP+PRS). I raise this only as a suggestion for the authors' consideration.

[LINK]

---

## [Decision Letter · Decision Letter 2]

18 Aug 2020

Dear Dr. Di Angelantonio,

Thank you very much for submitting your manuscript "Polygenic risk scores in cardiovascular risk prediction: 

prospective cohort study and modelling analyses" (PMEDICINE-D-20-00284R2) for consideration at PLOS Medicine. 

[LINK]

In light of these reviews, I am afraid that we will not be able to accept the manuscript for publication in the journal in its current form, but we would like to consider a revised version that addresses the reviewers' and editors' comments. Obviously we cannot make any decision about publication until we have seen the revised manuscript and your response, and we plan to seek re-review by one or more of the reviewers. 

We expect to receive your revised manuscript by Sep 08 2020 11:59PM. Please email us (plosmedicine@plos.org) if you have any questions or concerns.

We look forward to receiving your revised manuscript. 

Sincerely,

Clare Stone, PhD

Acting Chief Editor 

PLOS Medicine

plosmedicine.org

Please address all points from Referee 1 - note we will only consult with this referee one more time.

Comments from the reviewers:

Reviewer #1: Thanks authors for their effort to improve the manuscript and I am satisfied with some of the responses. However, there are still remaining issues needing attention.

1) Regarding response #2, the authors didn't respond to my question on the calibration of the developed models. As described in the TRIPOD statement, the performance of clinical prediction models consists of both discrimination and calibration. So far in the paper the C-index and NPI are all about discrimination. However, there is no calibration of the models in the paper such as calibration plot/slope and other measures such as RMSE (root mean square error), therefore the performance evaluation is only half done.

2) Regrading response #3, I am still not convinced about the clinical usefulness of adding the PRSs to conventional risk factors as only marginal improvement of around 1% in C-index was shown in the study which is also subject to complete independent external validation (eg, in another European country). Of course, if apply to large population, the 1% will always convert to some numbers but not sure whether it is cost effective and practical. Authors haven't touched the point of implication and practical usefulness of this marginal improvement by PRSs in the discussion, which is not adequate. Basically, one would wonder whether it's worth it to use all these genetic tests to gain only marginal improvement in prediction which doesn't seem very useful as compared to conventional risk factors.

Reviewer #2: The authors addressed all my concerns

[LINK]

---

## [Decision Letter · Decision Letter 3]

17 Nov 2020

Dear Dr. Di Angelantonio,

Thank you very much for re-submitting your manuscript "Polygenic risk scores in cardiovascular risk prediction: 

prospective cohort study and modelling analyses" (PMEDICINE-D-20-00284R3) for review by PLOS Medicine.

I have discussed the paper with my colleagues and the academic editor and it was also seen again by reviewers. I am pleased to say that provided the remaining editorial and production issues are dealt with we are planning to accept the paper for publication in the journal.

[LINK]

We look forward to receiving the revised manuscript by Nov 24 2020 11:59PM. 

Sincerely,

Adya Misra

Senior Editor

PLOS Medicine

plosmedicine.org

Requests from Editors:

Title- please reword to "Polygenic risk scores in cardiovascular risk prediction: a modelling study"

Abstract- please add brief participant demographics

Abstract -Might be helpful to describe the C-index?

The overall tone and language needs to be toned down to avoid overstating conclusions. Please add “our results indicate” or “modelling suggests” etc and adopt more cautious language

Conclusion- please add “our results suggest” or similar, to avoid overstating results. I’m not sure how this approach directly translates to application at scale, I suggest removing this part

Author summary- please add bullet points

Model assumptions mentioned on page 8 should be briefly mentioned in the abstract as well

Line 222, "to avoid optimism" should perhaps be "to critically evaluate this possibility ..." or similar?

Line 340-342 I suggest replacing this with a summary of the findings

Line 348- 20,000 CVD events prevented-- needs to be toned down as this is not directly tested in your study

Please remove financial information and data availability statements from the main text and add these to the article meta-data sections instead

Did your study have a prespecified protocol or analysis plan? Please state this (either way) early in the Methods section.

a) If a prespecified analysis plan (from your funding proposal, IRB or other ethics committee submission, study protocol, or other planning document written before analyzing the data) was used in designing the study, please include the relevant prospectively written document with your revised manuscript as a Supporting Information file to be published alongside your study, and cite it in the Methods section. A legend for this file should be included at the end of your manuscript.

TRIPOD checklist- please use paragraphs and sections instead of page numbers as these are likely to change

Please remove spaces from square brackets

Please ensure p-values are reported to up to three decimal spaces. For eg Fig4 contains - p<0.0001

Comments from Reviewers:

Reviewer #1: Many thanks authors for their great effort to improve the manuscript. I am satisfied with the response and revision. No further issues needing attention.

[LINK]

---

## [Editor Report · Decision Letter 4]

14 Dec 2020

Dear Dr. Di Angelantonio,

I am writing concerning your manuscript submitted to PLOS Medicine, entitled “Polygenic risk scores in cardiovascular risk prediction: 
a cohort study and modelling analyses.”

We have now completed our final technical checks and have approved your submission for publication. You will shortly receive a letter of formal acceptance from the editor.

Kind regards,

PLOS Medicine